# Solving a Real-Life Distributor's Pallet Loading Problem

**Mauro Dell'Amico**  **and Matteo Magnani** *

Department of Sciences and Methods for Engineering, University of Modena and Reggio Emilia, Via Amendola 2, 42122 Reggio Emilia, Italy; mauro.dellamico@unimore.it
* Correspondence: ma.magnani@unimore.it

**Abstract:** We consider the distributor's pallet loading problem where a set of different boxes are packed on the smallest number of pallets by satisfying a given set of constraints. In particular, we refer to a real-life environment where each pallet is loaded with a set of layers made of boxes, and both a stability constraint and a compression constraint must be respected. The stability requirement imposes the following: (a) to load at level $k + 1$ a layer with total area (i.e., the sum of the bottom faces' area of the boxes present in the layer) not exceeding $\alpha$ times the area of the layer of level $k$ (where $\alpha \geq 1$), and (b) to limit with a given threshold the difference between the highest and the lowest box of a layer. The compression constraint defines the maximum weight that each layer $k$ can sustain; hence, the total weight of the layers loaded over $k$ must not exceed that value. Some stability and compression constraints are considered in other works, but to our knowledge, none are defined as faced in a real-life problem. We present a matheuristic approach which works in two phases. In the first, a number of layers are defined using classical 2D bin packing algorithms, applied to a smart selection of boxes. In the second phase, the layers are packed on the minimum number of pallets by means of a specialized MILP model solved with Gurobi. Computational experiments on real-life instances are used to assess the effectiveness of the algorithm.

**Keywords:** distributor's pallet loading problem; heuristics; bin packing; real-life instances



## 1. Introduction

The *distributor's pallet loading problem* (DPLP) is a topic of wide interest for operational research and companies that deal with logistics, transport, and storage, in addition to production that leads to small and medium-sized packaging.

The problem is to find the optimal loading of parallelepiped-shaped boxes, not necessarily with the same sizes, on the fewest possible number of pallets, with predefined dimensions and weight limit. We consider the special case where the loading of each pallet is done by adding layers of boxes, one on top of the other. This case is very frequent in real-life applications.

Achieving the goal of minimizing the number of pallets built in an acceptable time means significantly reducing the costs due to the transport and storage of materials.

In practical problems, we have to deal with not only sizes but weights and the capacity of boxes to sustain other boxes. In this paper, we consider the following properties, which induce specific constraints:

- **stability**: that is, the property of a layer to sustain other layers, possibly with a larger area;
- **weight limit**: the sum of the weights of all boxes loaded on a pallet, which must not be greater than a certain limit given by the company;
- **compression limit**: capacity of a layer of boxes to support the weight of the boxes above it.

DPLP is strongly NP-hard, since it is a generalization of the *bin packing problem* (BPP) [1–6]. In the BPP we are given N items, each characterized by a weight, and an infinite number

of bins of a given capacity. The problem is to load all the items in the minimum number of bins by respecting the capacity constraint. It is easy to see that the DPLP can model the BPP by disregarding all constraints but the one referring to the total weight of a pallet (bin capacity).

Since the 1960s, researchers and companies have deeply investigated the problem of cutting-stock, which is very similar to the BPP. In fact, in the cutting-stock problem, it is necessary to find a way to cut pieces of material, not necessarily with the same shapes and sizes, minimizing waste [7,8].

For a classification of some types of packing problems, refer to [9,10]; in [11], a review of the possible constraints most commonly used for packing and cargo problems is presented.

The DPLP is a generalization of the manufacturer's pallet loading problem (MPLP) [12,13], which deals with the same objective function but loads identical boxes on pallets.

For discussions of problems with constraints deriving from real applications, similar to the one in this paper, we refer to [14–21]. In particular, Ancora et al. [14] works with a hybrid genetic strategy; in [15–18], the authors use, respectively, heuristics, greedy approach, the genetic and differential evolution algorithm, and the branch and bound way to solve the DPLP; Gzara et al. [19] exploits a layer-based column generation; and Ancora et al. [14] works with a hybrid genetic strategy.

From a mathematical-modelling point of view, there is not much research that deals with the packing problem nor with real constraints (weight limit, compression, and stackability) as seen in this treatise. Nonetheless, the problem of 3D packing is usually dealt with by creating 2D layers, subsequently stacked on top of each other.

For a variant of this problem, the so-called container loading problem, which differs in the fact that, in general, constraints limiting the possibilities of layers overlapping are not explicit, we refer to [20–29]. (ILP strategy [23], GRASP method [20,24,25], heuristic way [27], heuristic-genetic algorithm [28], heuristics and MILP method [26], layer-based greedy strategy [21]).

We present in this paper a two-step algorithm to solve the pallet construction problem, basing our tests on real commercial orders from a logistics company using automated robots for creating and managing pallets. For similar work we refer to [30], which introduces visibility and contiguity constraints.

## 2. Materials and Methods

We are given a set $\mathscr{B}$ of 3D *boxes* partitioned into *types* associated with boxes of identical size, weight, and compression index. More specifically, let $\mathscr{I}$ denote the set of box types, and let $n_i$ denote the number of identical boxes of type $i \in \mathscr{I}$ (with $\sum_{i \in \mathscr{I}} n_i = |\mathscr{B}|$). Each box of type $i$ has width $w_i \in \mathbb{Z}_+^*$, depth $d_i \in \mathbb{Z}_+^*$, and height $h_i \in \mathbb{Z}_+^*$. Given a box $j \in \mathscr{B}$, we will denote with $i(j)$ the type of box $j$. We are also given an arbitrarily large set $\mathscr{P}$ of identical pallets. Each pallet has a two-dimensional loading surface of width $W \in \mathbb{Z}_+^*$ and depth $D \in \mathbb{Z}_+^*$, which can be used to load boxes up to a maximum height $H \in \mathbb{Z}_+^*$. We assume that $w_i \leq W$, $d_i \leq D$, $h_i \leq H$, for each $i \in \mathscr{I}$. Moreover, each box of type $i$ has a *weight* $p_i$ and a *compression index* $c_i$ that will be used to define the maximum weight the box can support.

The problem requires assigning all the boxes to the pallets by restricting the loading to layered solutions. A feasible packing of boxes on a pallet can be decomposed into subsets of boxes each defining a layer, and the layers are loaded one over the other. More formally, a *layer* is a subset of boxes which can rotate 90 degrees on their support surface whose basis can be packed into a $W \times D$ rectangle.

Let us define as $\mathscr{L}$ the set of all the possible types of layers we could build with boxes $\mathscr{B}$. Observe that $\mathscr{L}$ has exponentially many elements, so we will adopt algorithms that only consider a subset of these layers. A layer $l \in \mathscr{L}$ is given by the set of boxes assigned to it, say $\mathscr{B}_l$, and by a specific 2D packing of these boxes (more precisely, of the basis of the boxes), whose total width must not exceed $W$, and whose total depth must not exceed

*D*. Let $H_l = \max_{j \in \mathscr{B}_l} h_{i(j)}$ denote the height of the layer, and let $A_l = \sum_{j \in B_l} h_{i(j)} d_{i(j)}$ denote the total area of the layer.

A layer built with boxes of the same height produces a planar surface (possibly with some holes), on which we can load another layer. However, the practical experience of the company has shown that it is not strictly necessary that the bottom layer has a perfectly planar upper surface to be used as support for another layer: it is enough that this surface is not too wavy. This can be translated into a simple requirement, imposing that the difference in height of its boxes is small enough. More precisely, given a layer *l*, we impose

$$|h_i - h_j| \leq \Delta h \quad \forall i, j \in \mathscr{B}_l, \tag{1}$$

in which parameter $\Delta h$ defines the maximum height difference allowed between two boxes of a layer. We say that a layer for which (1) holds satisfies the **stackability constraint**. We will consider this as a unique exception to the last layer of a pallet (top layer): since no other layer will be loaded on the top one, the restriction on the difference of heights does not have to be considered.

Another constraint for feasible loading of one layer over another regards the ratio of the total areas of the boxes in the layer. It is obvious that if we load a layer with a large area on a layer with a very small area, there is an issue with the stability of the overlying layer. The **stability constraint**, defined by the inequality

$$A_l \geq \alpha A_m, \tag{2}$$

requires loading layer *m* immediately over layer *l*, where $\alpha \geq 1$ is a given parameter.

To each layer *l*, an overall *compression factor* is also associated:

$$C_l = \begin{cases} \min_{j \in \mathscr{B}_l} c_{i(j)} A_l & \text{if } l \text{ satisfies (1)} \\ 0 & \text{otherwise} \end{cases} \tag{3}$$

giving the maximum total weight the layer can support. We will call this requirement the **compression constraint** (see [14,18,19,21] for other studies that consider this constraint in a similar way). Note that giving a zero compression factor to layers not satisfying (1) implies that these layers can be only packed as the top layer of a pallet.

Resuming the above descriptions, the aim of the DPLP that we face is to load all boxes into the minimum number of pallets by ensuring that the following constraints are satisfied:

c1. Numerosity constraint: all boxes in $\mathscr{B}$ must be packed;
c2. Height constraint: the sum of the heights $H_l$ of all layers loaded on a pallet must not exceed *H*;
c3. Stackability constraint: each layer, except the top one of each pallet, must be composed by boxes satisfying (1);
c4. Stability constraint: each pair of layers *m, l*, with *m* loaded immediately over *l*, must satisfy (2);
c5. Compression constraint: the total weight of all boxes in the layers loaded over a layer *l* cannot exceed the compression factor $C_l$.

## 2.1. Creating 2D Layers

For the creation of the layers' set $\mathcal{L}$, we again use a two-step method. In the first step, we partition the boxes into *families* of boxes with the procedure CREATEFAMILIES, which takes as input the number of families *f* in which we want to divide the set of box types and returns a partition $\mathscr{I}(1), \ldots, \mathscr{I}(f)$, where each family $\mathscr{I}(i)$ contains box types whose heights differentiate at most by $(1 + \gamma)\Delta h$, and $\gamma$ is a randomly selected small value. Therefore, we say that a family *almost* satisfies requirement **c3** (see (1)). In the second step, we apply to these families 2D packing methods from the literature to create the layers.

Our algorithm BUILDLAYERS of Algorithm 1 uses procedure CREATEFAMILIES (see Algorithm 2) and a set of heuristic algorithms 2D-$H_1, \ldots,$ 2D-$H_{a_{\max}}$.

---

**Algorithm 1:** Algorithm for creating the layers.

---

**Algorithm** BUILDLAYERS()
**input**: boxes $\mathcal{B}$ and their types $\mathcal{I}$, pallets' sizes
1.  Sort the box types in $\mathcal{I}$ by non decreasing heights (i.e., $h_1 \le h_2 \le \ldots, \le h_{|\mathcal{I}|}$)
2.  $\mathcal{L} = \varnothing$;
3.  **for** $f = f_{\min}$ **to** $f_{\max}$ **do**
4.  $\quad (\mathcal{I}(1), \ldots, \mathcal{I}(f)) = \text{CREATEFAMILIES}(f)$;
5.  $\quad$ **for** $i = 1$ **to** $f$ **do**
6.  $\qquad$ **for** $a = 1$ **to** $a_{\max}$ **do**
7.  $\qquad\quad$ pack the boxes in $\mathcal{I}(i)$ with heuristic 2D-$H_a$ giving layers $L$
8.  $\qquad\quad \mathcal{L} = \mathcal{L} \cup L$;
9.  $\qquad$ **endfor**
10. $\quad$ **endfor**
11. **endfor**
**return** $\mathcal{L}$

---

**Algorithm 2:** Procedure for creating the box types families.

---

**Procedure** CREATEFAMILIES(number of families $f$)
1.  Let $n = |\mathcal{I}|$
2.  Choose randomly $\gamma \in [0, 3; 0, 5]$
3.  **for** $j = 1, \ldots, f$ **do** let $\nu(j) = \text{RANDOM}\,((j-1)\lceil n/f \rceil, \min(j\lceil n/f \rceil, |\mathcal{I}|))$
4.  $\mathcal{I}(1) = \{1, \ldots, \bar{\iota}\}$ with $\bar{\iota} = \arg\max\{h_i \le h_{\nu(1)} + \frac{1}{2}(1+\gamma)\Delta h\}$
5.  **for** $j = 2$ **to** $f - 1$ **do**
6.  $\quad \mathcal{I}(j) = \{\underline{\iota}, \ldots, \bar{\iota}\}$ with $\underline{\iota} = \arg\min\{h_i \ge h_{\nu(j)} - \frac{1}{2}(1+\gamma)\Delta h\}$
    $\qquad \bar{\iota} = \arg\max\{h_i \le h_{\nu(j)} + \frac{1}{2}(1+\gamma)\Delta h\}$
7.  **endfor**
8.  $\mathcal{I}(f) = \{\underline{\iota}, \ldots, n\}$ with $\underline{\iota} = \arg\min\{h_i \ge h_{\nu(|\mathcal{I}|)} - \frac{1}{2}(1+\gamma)\Delta h\}$
9.  **for** $j = 1$ **to** $f - 1$ **do**
10. $\quad$ **if** $\mathcal{I}(j) \cap \mathcal{I}(j+1) \ne \varnothing$ **then**
11. $\qquad \mathcal{I}(j+1) = \mathcal{I}(j+1) \setminus \mathcal{I}(j)$
12. $\quad$ **endif**
13. **endfor**
14. **foreach** $i \in \mathcal{I}$ **do**
15. $\quad$ **if** $i \notin \cup_{j=1}^{f}\mathcal{I}(j)$ **then**
16. $\qquad$ assign $i$ to the set with nearest pivot height
17. $\quad$ **endif**
18. **endfor**
**return**$(\mathcal{I}(1), \ldots, \mathcal{I}(f))$

---

The loop 'for' at line 3 of algorithm BUILDLAYERS calls CREATEFAMILIES a few times with different values of the partition's number, $f$, to create different families. At each family, the 2D packing heuristics 2D-$H_1, \ldots,$ 2D-$H_{a_{\max}}$ are applied in turn, which produces layers that are added to the layer set $\mathcal{L}$. Note that due to the 'almost' satisfaction of requirement **c3**, set $\mathcal{L}$ will contain both layers satisfying (1) or not.

Procedure CREATEFAMILIES starts by dividing the interval of the box types into $f$ subinterval of identical length, except the last one, which could have, in some cases, fewer elements than the other subinterval, and selects a pivot box type index $\nu(i)$ from each subinterval $i$ (with $i \in \{1, \ldots, f\}$). Each family $\mathcal{I}(i)$ is defined as the set of box types with absolute height difference from $h_{\nu(i)}$ not exceeding $\frac{1}{2}(1+\gamma)\Delta h$. Given this first selection of box types, the possible intersections are then removed, and the box types not inserted into any subset, if any, are assigned to the set with closest pivot height.

The need to insert the random parameter $\gamma$ and to choose the random pivot derives from the fact that we want to create slightly different families at each iteration. By doing so, we provide the possibility of creating different layers at each iteration, keeping the families unchanged for most of the elements (because $\gamma$ is small).

For the same reason, in our experiments, we used some 2D packing methods from [31], named the maximal rectangle and skyline algorithms.

In particular, the maximal rectangle algorithm works by storing a list of free rectangles, which represent the free area of the layer. Every time a rectangle is placed in a layer (if possible, otherwise a new one is opened), the list of free rectangles of that layer is updated, adding 2 rectangles that form an L-shaped region as shown in Figure 1.

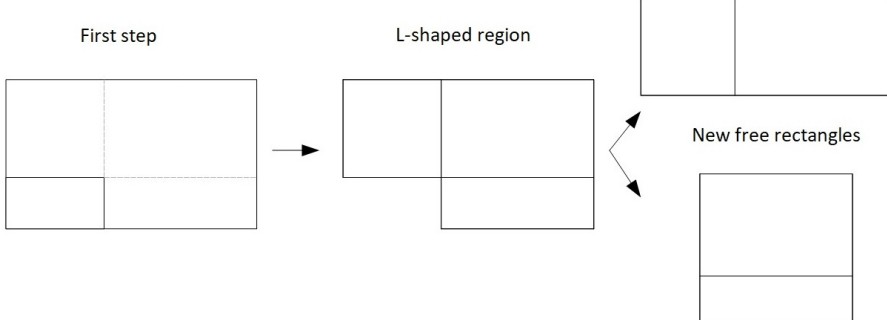

**Figure 1.** Free Rectangles.

Any overlaps between free rectangles or degenerate free rectangles are eliminated each time the list of free rectangles is updated.

The skyline structure is the same as that of the *envelope* in [32]: a list containing the high edges of the already packed rectangles is updated every time a new rectangle is placed in the current layer (if possible, otherwise a new one is opened). Due to the simplicity of the structure at the base of the skyline, it is easy to memorize the areas that would otherwise no longer be used during packing (any holes in the packing); let us call these areas waste map. Before looking for new locations for the new rectangles, we try to place them on the waste map.

For both structures, we used different approaches for choosing where to place the rectangles:

- MaxRectBL: maximal rectangle with bottom-left strategy (place each rectangle in the position where the y-coordinate of the top side of the rectangle is the smallest, and if there are several such valid positions, pick the one that has the smallest x-coordinate value);
- MaxRectBLR: maximal rectangle with bottom-left strategy and rotation allowed;
- MaxRectBssfR: maximal rectangles best short side fit strategy chooses to pack the current rectangle into the free rectangle, which minimizes the differences between the dimensions of the rectangle and the free one;
- SkylineBlWm: skyline with bottom-left and waste map strategy;
- SkylineBlWmR: skyline with bottom-left and waste map strategy with rotation allowed;
- SkylineMwfWm: skyline with min waste fit with low profile heuristic, minimizing the area wasted below the rectangle; at the same time, it tries to keep the height minimal;
- SkylineMwfWmR: skyline with min waste fit with low profile heuristic and rotation allowed.

### 2.2. A Mathematical Model for Loading Layers

In this section, we present a mathematical model for the optimal loading of layers in the minimum number of pallets.

We represent the layers' types obtained in the first phase by a $|\mathscr{I}| \times |\mathcal{L}|$ matrix $A$ where $a_{il}$ denotes the number of boxes of type $i$ packed in layer $l$.

We will use two sets of binary variables. Variable $y_p$ will take value 1 if the pallet $p \in \mathscr{P}$ is used, and zero otherwise. Variable $x_{lpk}$ will have value 1 if layer $l$ is stacked on pallet $p$ at level $k$, and 0 otherwise.

Parameter $\kappa = \lfloor H/min_{i \in \mathscr{I}} h_i \rfloor$ denotes the maximum number of layers that can be loaded on any pallet.

Finally, we calculate the weight of each layer as the sum of the weights of the items present on each one:

$$P_l = \sum_{i \in \mathscr{I}} a_{il} p_i$$

Then, the mathematical model for stacking on pallets is as follows:

$$\min \sum_{p \in \mathscr{P}} y_p \tag{4}$$

$$\sum_{k=1}^{\kappa} \sum_{p \in \mathscr{P}} \sum_{l \in \mathcal{L}} a_{il} x_{lpk} = n_i \qquad\qquad i \in \mathscr{I} \tag{5}$$

$$\sum_{k=1}^{\kappa} \sum_{l \in \mathcal{L}} H_l x_{lpk} \le H y_p \qquad\qquad p \in \mathscr{P} \tag{6}$$

$$\sum_{k=1}^{\kappa} \sum_{l \in \mathcal{L}} P_l x_{lpk} \le P y_p \qquad\qquad p \in \mathscr{P} \tag{7}$$

$$x_{lpk} + x_{mp(k+1)} \le 1 \qquad\qquad l, m \in \mathcal{L}, : A_l \ge \alpha A_m, \ p \in \mathscr{P}$$
$$k \in \{1, \dots, \kappa - 1\} \tag{8}$$

$$\sum_{h=k+1}^{\kappa} \sum_{\ell \in \mathcal{L}} x_{\ell ph} P_\ell \le C_l + M(1 - x_{lpk}) \qquad l \in \mathcal{L}, p \in \mathscr{P}, k \in \{1, \dots, \kappa - 1\} \tag{9}$$

$$\sum_{l \in \mathcal{L}} x_{lpk} \le 1 \qquad\qquad p \in \mathscr{P}, k \in \{1, \dots, \kappa\} \tag{10}$$

$$\sum_{l \in \mathcal{L}} (x_{lpk} - x_{lp(k-1)}) \le 0 \qquad\qquad k \in \{2, \dots, \kappa\}, p \in \mathscr{P} \tag{11}$$

$$y_p \le y_{p-1} \qquad\qquad p \in \{2, \dots, |\mathscr{P}|\} \tag{12}$$

$$y_p \in \{0, 1\} \qquad\qquad p \in \mathscr{P} \tag{13}$$

$$x_{lpk} \in \{0, 1\} \qquad\qquad l \in \mathcal{L}, p \in \mathscr{P}, k \in \{1, \dots, \kappa\} \tag{14}$$

The objective function (4) minimizes the number of pallets used. Constraint (5) requires that each box is packed once on the pallet, thus implementing requirement **c1**. Constraints (6) implement requirement **c2** by imposing that the sum of the heights of the layers on each pallet does not exceed the available height $H$. Constraints (7) require that the sum of the weights of the boxes on each pallet do not exceed $P$, where $P$ is the maximum weight that can be loaded on each pallet. The stability requirement **c4** is satisfied by constraints (8), while the compression requirement **c5** is guaranteed by (9) ($M$ being, as usual, a very large positive number). Note that requirement **c3** is satisfied by definition (3) for all layers with $C_l > 0$, while when $C_l = 0$, constraints (9) impose that the layer be packed as the top one of a pallet.

To conclude, the model constraints (10), (11), and (12) respectively impose the following: (a) to load at most one layer per level, (b) to load a level $k$ only if level $k-1$ has been loaded, and (c) to use a pallet $p$ only if pallet $p-1$ has been used. The definition of the domains of the variables follows.

## 3. Results

All experiments have been conducted on a PC with Intel Core i7-10510U CPU 2.30 GHz, 16 GB RAM, and Windows 10 Operating System. The algorithms have been implemented in Python 3.8 and run using PyCharm 2021.1.2.

We solved the MILP model with Gurobi 9.1.1. We considered a set of five instances from real orders within the corresponding company manual solutions. For each instance, we set the time limit to 120 min for the Gurobi solver, running the algorithm five times, using all variants of the skyline and maximal rectangle algorithms we presented in Section 2.1.

For all instances, the pallet dimensions were set to 1650, 1200, and 800 for height, width, and length, respectively. The maximum weight $P$ loadable on every pallet was set to 4000, and the $\Delta h$ was set to 20 for every layer in $\mathcal{L}$. Finally, $\alpha$ in (2) was set to 1.1.

In the following Tables 1–7, we show some experimental results, all based on real commercial orders of a logistics company.

**Table 1.** Instance details.

| Instance | N° Items | N° Items Type | Tot. Weight | Min. Compr. | Max. Height | Min. Height |
|:---:|:---:|:---:|:---:|:---:|:---:|:---:|
| A | 332 | 53 | 2967.58 | 87.5 | 305 | 150 |
| B | 136 | 22 | 1564.56 | 87.5 | 305 | 150 |
| C | 349 | 70 | 3272.756 | 87.5 | 305 | 150 |
| D | 669 | 68 | 6901.96 | 87.5 | 305 | 150 |
| E | 83 | 14 | 464.83 | 75 | 265 | 150 |

**Table 2.** First run for all instances.

| Instance | $|\mathcal{L}|$ | Best Bound | Best Solution | Comp. Solution | Time (min) |
|:---:|:---:|:---:|:---:|:---:|:---:|
| A | 139 | 5 | 6 | 7 | 120 |
| B | 65 | 3 | 3 | 4 | 3 |
| C | 180 | 7 | 8 | 8 | 120 |
| D | 220 | 11 | 12 | 13 | 120 |
| E | 34 | 1 | 1 | 2 | 2 |

**Table 3.** Second run for all instances.

| Instance | $|\mathcal{L}|$ | Best Bound | Best Solution | Comp. Solution | Time (min) |
|:---:|:---:|:---:|:---:|:---:|:---:|
| A | 141 | 5 | 6 | 7 | 120 |
| B | 68 | 3 | 3 | 4 | 4 |
| C | 174 | 7 | 9 | 8 | 120 |
| D | 228 | 11 | 12 | 13 | 120 |
| E | 34 | 1 | 1 | 2 | 2 |

**Table 4.** Third run for all instances.

| Instance | $|\mathcal{L}|$ | Best Bound | Best Solution | Comp. Solution | Time (min) |
|:---:|:---:|:---:|:---:|:---:|:---:|
| A | 136 | 5 | 6 | 7 | 120 |
| B | 67 | 3 | 3 | 4 | 3 |
| C | 178 | 7 | 8 | 8 | 120 |
| D | 222 | 11 | 12 | 13 | 120 |
| E | 32 | 1 | 1 | 2 | 2 |

**Table 5.** Fourth run for all instances.

| Instance | $|\mathcal{L}|$ | Best Bound | Best Solution | Comp. Solution | Time (min) |
|:---:|:---:|:---:|:---:|:---:|:---:|
| A | 132 | 5 | 7 | 7 | 120 |
| B | 66 | 3 | 3 | 4 | 3 |
| C | 174 | 7 | 8 | 8 | 120 |
| D | 229 | 11 | 12 | 13 | 120 |
| E | 33 | 1 | 1 | 2 | 2 |

**Table 6.** Fifth run for all instances.

| Instance | $|\mathcal{L}|$ | Best Bound | Best Solution | Comp. Solution | Time (min) |
|:---:|:---:|:---:|:---:|:---:|:---:|
| A | 137 | 5 | 6 | 7 | 120 |
| B | 64 | 3 | 3 | 4 | 3 |
| C | 174 | 7 | 8 | 8 | 120 |
| D | 231 | 11 | 12 | 13 | 120 |
| E | 32 | 1 | 1 | 2 | 2 |

**Table 7.** Average of computational results.

| Instance | $|\mathcal{L}|$ | Best Bound | Best Solution | Comp. Solution | Time (min) |
|:---:|:---:|:---:|:---:|:---:|:---:|
| A | 137 | 5 | 6 | 7 | 120 |
| B | 66 | 3 | 3 | 4 | 3 |
| C | 176 | 7 | 8 | 8 | 120 |
| D | 226 | 11 | 12 | 13 | 120 |
| E | 33 | 1 | 1 | 2 | 2 |

We ran the algorithm BUILDLAYERS $F$ times, where $F = f_{\max} - f_{\min} + 1$, as the creation of the layers based on families is partially randomized. In particular, we set $f_{\min} = 2$ and $f_{\max} = |\mathcal{H}|$, where $\mathcal{H}$ is the set of different heights of boxes which are present in the commercial order. The use of multiple layerization methods, multiple division into families, and partial randomization allows for the creation of different layers that can expand the pool of layers (see Figure 2 for our results). It is possible that by adding some layers to $\mathcal{L}$, even if they are slightly different from each other, the general solution to the problem is greatly improved. The algorithm always improves the manual solutions, on average. In very few cases (2 out of 25), the proposed solution equals the manual one, but never exceeds it (see Figure 3).

The time columns refer to the time required by Gurobi to solve the problem. Small commercial orders (Instances B and E) are processed in a short time, as few layers are created, making the minimum convergence of the palletizing model fast. On the other hand, orders that have many boxes and many box types (Instances A, C, and D) require more computing time, often reaching the time limit.

These computing times are compatible with the practical management of large orders when loading can be planned early. From the computational point of view, the most onerous process is represented by the resolution of the palletization model. In fact, even for the largest orders, the filling of the layers' pool ends in a maximum of 5 min.

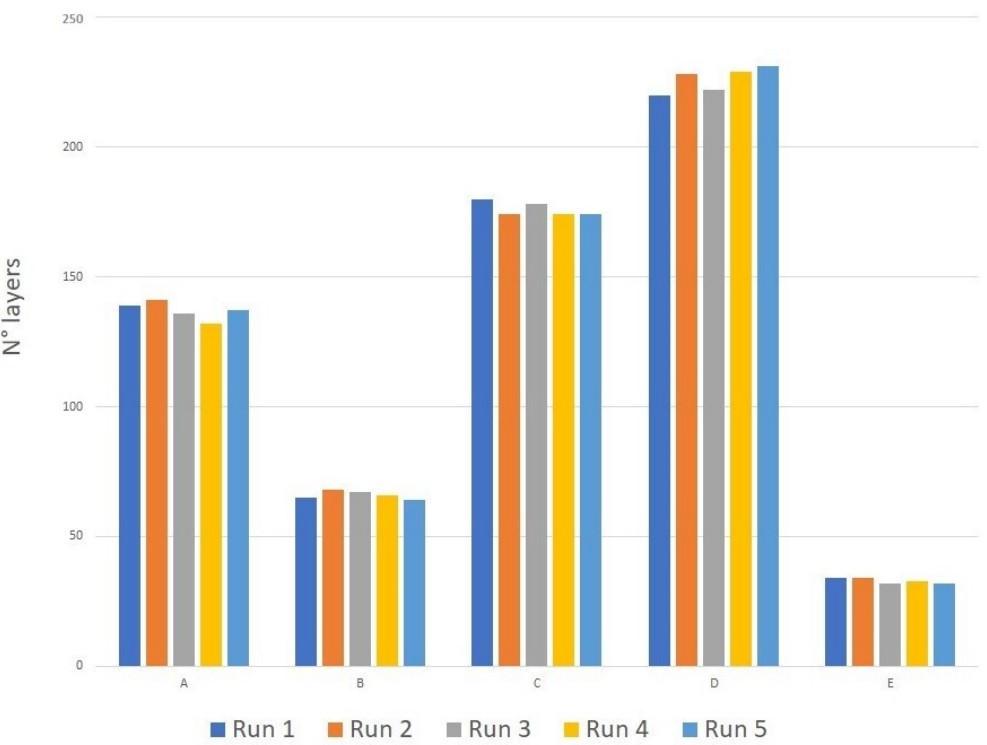

**Figure 2.** Layer for every run, for every instance.

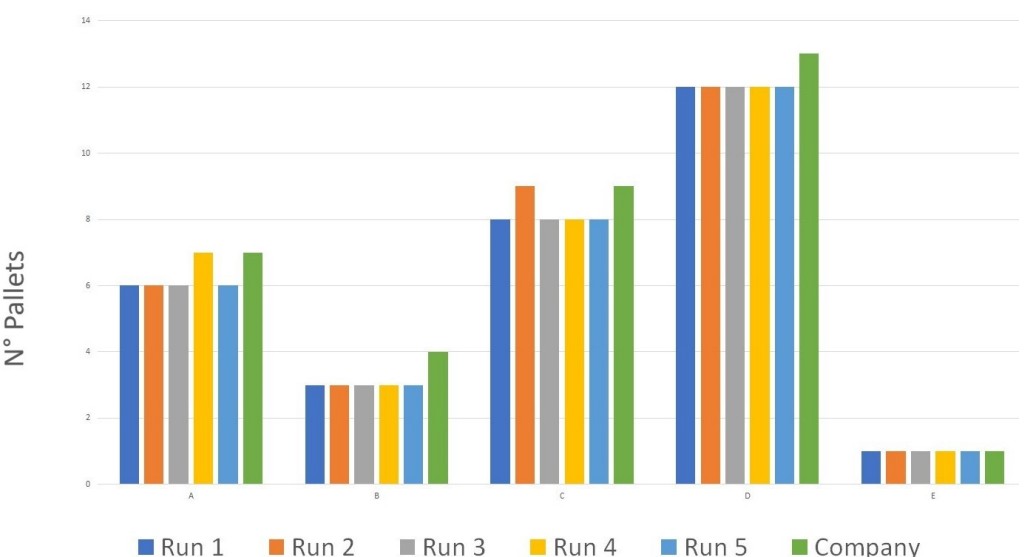

**Figure 3.** Pallets for every run, for every instance, with company results.

For the sake of privacy required by the company, we are not able to publish the complete database we used for the experiments.

We want to highlight that this algorithm has some advantages: it solves a real problem with constraints deriving from real experiences; it is possible to obtain substantial modifications: for example, changing the algorithm that solves the 2D layer creation problem or varying the $\gamma$ hyperparameter of the families' creation. It can be sped up by dividing into fewer families or by using fewer 2D packing algorithms.

We are confident that in the future, with experiments on more orders and with changes to the hyperparameters, this algorithm can achieve even more satisfactory results than those obtained in this paper.

**Author Contributions:** Conceptualization, M.M. and M.D.; methodology, M.M. and M.D.; software, M.M.; validation, M.M. and M.D.; formal analysis, M.M. and M.D.; investigation, M.M.; resources, M.M.; data curation, M.M.; writing—original draft preparation, M.M.; writing—review and editing, M.D.; visualization, M.M.; supervision, M.D.; project administration, M.M. All authors have read and agreed to the published version of the manuscript.

**Funding:** This research received no external funding.

**Conflicts of Interest:** The authors declare no conflict of interest.

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
