# Peer review of "Solving a Real-Life Distributor’s Pallet Loading Problem"

_mca, doi:10.3390/mca26030053_

Round 1

Reviewer 1 Report

Comments are listed below (file attached).

Author Response

ANSWER TO REFEREE number 1

We would like to thank the reviewer for his/her suggestions that helped us improve the paper.

The manuscript describes a matheuristic algorithm to address the Distributor’s Pallet Loading Problem, by satisfying both stability and compression constraints. The authors decompose the problem into two parts, solving the 2D version by using an adaptation of heuristic algorithms from the literature, and then solving the 1D version through a MILP model. The problem is well defined and described.

The paper presents a formal mathematical description for pallet creation. The performance of the algorithm is assessed through computational tests on real-world instances, although there is neither a discussion about the algorithm performance nor conclusions.

General evaluation

The paper is interesting from an industrial point of view because the authors address known constraints in an interesting manner. The topic of the paper is relevant and fits with the journal scope. The references are reasonably complete. Most of the parts of the paper is easy to follow and understand.

To sum up, the presented contribution is sufficient for publication after a major revision.

Thank you for your appreciation.

  • Abstract

Line 6: ‘i.e. the sum of the areas of the boxes in the layer’. Addressing a Pallet Loading Problem, we could say ‘area of the box’ after indicating which of its face is used (bottom face, for example).

Line 9: ‘k must not exceed this value’. I suggest using ‘that’ instead of ‘this’.

Line 11: ‘our real-life problem’. I think it is more formal saying ‘real-life problem faced’ or ‘real-life problem addressed’.

We followed your suggestions, correcting the text.

  • Introduction

It also contains the literature review. In this case, the authors should make an effort in explaining in more details the problem, for instance, including practical examples of applications. Regarding the literature review, they should include more references related to the addressed problem in addition to briefly explain the algorithmic methods to solve it. Finally, they could avoid rewriting their ideas in the same Section, for example, paragraphs starting at lines 38 and 69. Also, line 65 could be modified to this purpose.

Line 20: ‘for companies who deal with’. I suggest using ‘that’ instead of ‘who’.

Check the reference at line 26. The authors include [1-5], but the correct reference to 3D Bin Packing might be [1-4]

Paragraph starting at line 44: they must include references in both sentences.

We modified the introduction and corrected mistakes and implemented your suggestions.

  • The concept of family is used at line 104 before it is defined.

We corrected this part, by removing the reference to families (which is not needed) in this statement.

  • Line 108. The stability constraint is not an equation, but it is a constraint defined by the equation (2). Therefore, I suggest including ‘defined by the inequality…’ or a similar phrase.

Corrected as suggested.

  • Line 110. Compression constraint is a simplification/approximation of this concept. The authors could include this characteristic in the sentence and cite some studies that also consider this constraint.

We cited some studies in Section 2, page 3, lines 111-113.

Line 128: ‘whose types almost satisfy (1)’. The explanation of ‘almost’ is included at line 135. I think it could be written in the same line (128), or the authors should indicate that it will be described below.

We corrected this part, now the statement is “In the first step we partition the boxes into families of boxes with the procedure CREATEFAMILIES, which takes as input the number of families f in which we want to divide the set of box types, and returns a partition I (1), . . . ,I ( f ), where each family I (i) contains box types whose heights differentiate at most by (1 + g)Dh and g is a randomly selected small value. Therefore we say that a family almost satisfy requirement c3

  • Line 144: ‘except the last one’. The last interval will always be lower than the other ones? Is not ‘except’ too limiting?

We added the phrase “which could have, in some cases, less elements then the other subinterval”. The cardinality of the last subinterval depends on the cardinality of the whole interval (if it is, or not, divisible for an integer which varies from f_min to f_max)

  • Line 145: in the set representation there is not a start value.

We corrected this oversight.

  • Procedure CreateFamilies

Line 1: [0,3,0,5] it this an interval?

Line 1: there are two different commands in the same line. It could be better writing them in separated lines.

Line 10: the word ‘exclude’ is misspelling, although I think it is an unnecessary comment in the pseudo-code.

Also, I think the most important steps that describes the pseudo-code should be included in the paragraph that starts at line 143.

We  implemented all suggestions. We believe now the pseudocode is more readable.

  • Line 149: the authors use some variants of the 2D packing method described in [27]. Since it is a variation, they should explain which variation/modification they applied and its main idea.

We included in our paper the description of the 2D packing algorithms, with the variations we utilized.

  • Line 151: ‘rotated by 90 degrees’. When introducing the information on the items, it should be explicitly stated that a 90 degree rotation is allowed.

We wrote this information in page 2, line 84.

  • MILP model on page 5

Constraint 8: parameter P is used, but it was not defined previously.

Constraint 13: variable ‘x’ instead of ‘k’.

  • Line 172: ‘Conclude the model constraints (9) and (11)’. It should be ‘Conclude the model constraints (9), (10), and (11)’.

  • Line 182: ‘LWF algorithm we talked about in the previous section’. The referred algorithm was not introduced/explained previously (I also think ‘talked about’ is not formal for a paper). Also, it should be referred correctly (previous section or Section 2).

  • Line 185: in that line alpha is defined to be equal to 0.9, but in Section 2 it was defined alpha >= 1.

All these mistakes were fixed. Thank you for spotting out.

  • Line 186: is the database available? It would be interesting to include it on an online repository, if available. If not, the authors should explain the reason it is not available (privacy, maybe).

Unfortunately the company does not allow to make public its data. We added a statement in the paper at page 11, lines 240-241.

  • Tables 2-6 are not necessary in my opinion, unless each one is for a variant of the LWF algorithm. The respective tables could be included on a personal repository and referred in the manuscript as Appendix.

We believe that the tables add some information, but if the editor prefers to move them to the Appendix we have no objection.

  • Line 193. ‘is better than that obtained from the business algorithm.’ To make a comparison we must present the related values.

Thanks, it is a good suggestion. We added to the histogram the company results.

  • Discussion of the results

There is no discussion of the results. It should be included in order to highlight the strong/weak parts of the full algorithm, as well as the advantages/disadvantages for each variant.

We added a brief discussion below the histogram.

  • Conclusion

Should be included.

Conclusions are not required by the journal policy, but we added a short comment, as you suggested.

  • References

There are references: missing information (15); incorrect paper name (10); not following the bold pattern for date (2,9,14,15,24,26). Should be checked.

We checked and corrected the mistakes on references.

The bold pattern was corrected by the Journal, so we think we have to respect their standard.

Reviewer 2 Report

The paper describes a variant of the palleting problem called the Distributor's Pallets Loading problem.

In this variant, there are a number of new constraints which are not part of the standard palleting problem.

 They have proposed some heuristics and compared their solution against the solutions produced by a solver.

  I have the following concerns:

 (a) The computational complexity of the problem is not even mentioned, much less discussed.

 (b) There is no analysis of their proposed heuristic. Why does it perform the way it does? What insights were involved in the development of the heuristic?

 (c) They claim that their computational experiments were carried out on real-life orders of a commercial company. While I can see this as a consulting effort, it is not clear that this is a research effort.

 (d) The English is absolutely atrocious. There is hardly a paragraph without serious grammatical or typographical errors.

 (e) It is unclear where the constraints c1-c5 arose from.

 (f) The problem description is confusing at best.

 What I got from this paper is that they have taken a standard problem, overloaded it with a bunch of constraints and then supplied it to a solver.

 I am not very confident that this is publishable material.

Author Response

ANSWER TO REFEREE number 2

We want to thank the referee for his/her valuable comment that helped us to improve the paper.

  • The computational complexity of the problem is not even mentioned, much less discussed.

We added a description of the computational complexity at page 1-2, lines 37-42.

  • There is no analysis of their proposed heuristic. Why does it perform the way it does? What insights were involved in the development of the heuristic?

We included in our paper a brief  description of the computational performances at page 10, lines 225-239.

  • They claim that their computational experiments were carried out on real-life orders of a commercial company. While I can see this as a consulting effort, it is not clear that this is a research effort.

The problem was suggested by a company, for the subject of a Master thesis. We realized that similar problems have been studied in the literature, but not the one here reported. Therefore we developed an original research modeling the new constraints and requirements, and testing the resulting model on the company instances.

  • The English is absolutely atrocious. There is hardly a paragraph without serious grammatical or typographical errors.

 We are really sorry for this. We have now carefully revised the manuscript and required an external mother tongue reader to check the current version.

  • It is unclear where the constraints c1-c5 arose from.

We have improved the introduction and description of the problem. We believe that now the constrains can be seen as the natural formal representation of the previous description.

  • The problem description is confusing at best.

We rephrased and corrected it, hoping that now it is more comprehensible.

What I got from this paper is that they have taken a standard problem, overloaded it with a bunch of constraints and then supplied it to a solver.

We took a real problem based on a company issue. We carefully looked in the  existing literature for the same problem, but  we didn’t found anything. We have thus developed an original modeling of the full problem. We have designed an heuristic approach to solve the problem and we used a solver for the second phase of this heuristic. We hope that with the new description it is more clear our contribution.

Round 2

Reviewer 1 Report

Comments are listed in the attached file.

Author Response

We want to thank the referee for his/her valuable comment that helped us to improve the paper.

Reviewer 2 Report

I have read the responses to my original contents and gone through the revised version of the paper.

 Here are my comments:

 (a) As far as I can see, they have used integer programming to model a non-traditional variant of bin-packing. The new variant involves a number of additional constraints, which can be captured by linear constraints using integer variables.

 (b) They have solved several instances using both their heuristics and Gurobi. The data needs to be explained in better fashion. It is not clear if Time in the final column refers to the time taken by Gurobi or their heuristic.

 (c) There needs to be a more detailed discussion of the empirical results. For instance, in most cases, the heuristic solution is off by at most one from the true optimal. Why is this? Can some bound be derived?

 (d) The exposition in the paper has improved significantly.

 (e) Clearly, I am not thrilled with the significance of the work. However, I am going to recommend acceptance, because there may be some merit in this work.

Author Response

(The authors gave the same response as above.)
